# Deep Exploration by Novelty-pursuit with Maximum State Entropy

## Abstract

Efficient exploration is essential to reinforcement learning in huge state space. Recent approaches to address this issue include the intrinsically motivated goal exploration process (IMGEP) and the maximum state entropy exploration (MSEE). In this paper, we disclose that goal-conditioned exploration behaviors in IMGEP can also maximize the state entropy, which bridges the IMGEP and the MSEE. From this connection, we propose a maximum entropy criterion for goal selection in goal-conditioned exploration, which results in the new exploration method *novelty-pursuit*. Novelty-pursuit performs the exploration in two stages: first, it selects a goal for the goal-conditioned exploration policy to reach the boundary of the explored region; then, it takes random actions to explore the non-explored region. We demonstrate the effectiveness of the proposed method in environments from simple maze environments, Mujoco tasks, to the long-horizon video game of SuperMarioBros. Experiment results show that the proposed method outperforms the state-of-the-art approaches that use curiosity-driven exploration.

## 1 Introduction

Efficient exploration is important to learn a (near-) optimal policy for reinforcement learning (RL) in huge state space (Sutton & Barto, 1998). Dithering strategies like epsilon-greedy, Gaussian action noise, and Boltzmann exploration are inefficient and require exponential interactions to explore the whole state space. In contrast, deep exploration (Osband et al., 2016) overcomes this dilemma via temporally extended behaviors with a long-term vision. Recently, proposed methods include the intrinsically motivated goal exploration process (IMGEP) (Forestier et al., 2017), and maximum state entropy exploration (MSEE) (Hazan et al., 2019). In particular, IMGEP selects interesting states from the experience buffer as goals for a goal-conditioned exploration policy. In this way, exploration behaviors are naturally temporally-extended via accomplishing self-generated goals. On the other hand, MSEE aims to search for a policy such that it maximizes the entropy of state distribution. In this way, the agent can escape from the local optimum caused by insufficient state exploration.

In this paper, we show that the target of maximizing the support of state distribution (discovering new states) and maximizing the entropy of state distribution (unifying visited state distribution) can be both achieved by the goal-conditioned policy. From this connection, we propose an exploration method called *novelty-pursuit*. Abstractly, our method performs in two stages: first, it selects a visited state with the least visitation counts as the goal to reach the boundary of the explored region; then, it takes random actions to explore the non-explored region. An illustration can be seen in Figure 1. Intuitively, this process is efficient since the agent avoids exploring within the explored region. Besides, the exploration boundary will be expanded further as more and more new states are discovered. Finally, the agent will probably explore the whole state space to find the optimal policy.

A naive implementation of the above strategies can lead to inefficient exploration and exploitation on complex environments. First, to tackle the problem of the curse of dimension and exhaustive storage when selecting the least visited states, we approximate the visitation counts via prediction errors given by Random Network Distillation (Burda et al., 2019b). Besides, we observe that previous methods used in IMGEP (Forestier et al., 2017) are inefficient to train the goal-conditioned exploration policy. We employ training techniques based on reward shaping (Ng et al., 1999) and HER (Andrychowicz et al., 2017) to accelerate training the goal-conditioned policy. Finally, we additionally train an unconditioned exploitation policy to utilize samples collected by the goal-conditioned

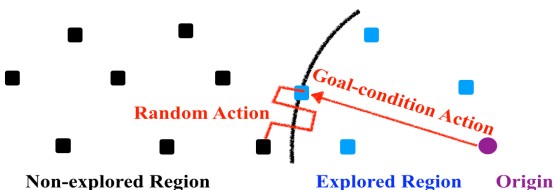

Figure 1: Illustration for the proposed method. A goal-conditioned policy firstly reaches the exploration boundary, then perform random actions to discover new states.

exploration policy with environment-specific rewards. Thus, the exploration and exploitation is decoupled in our method.

Our contributions are summarized as follows: (1) We disclose that goal-conditioned behaviors can also maximize the state entropy, which bridges the intrinsically motivated goal exploration process and the maximum state entropy explore. (2) We propose a method called novelty-pursuit from this connection and give practical implementations. (3) We demonstrate the exploration efficiency of the proposed method and achieve better performance on environments from the maze, Mujoco tasks, to long-horizon video games of SuperMarioBros.

## 2 BACKGROUND

**Reinforcement Learning.** In the standard reinforcement learning framework (Sutton & Barto, 1998) a learning agent interacts with a Markov Decision Process (MDP). The sequential decision process is characterized as follows: at each time $t$, the agent receives a state $s_t$ from the environment and selects an action $a_t$ from its policy $\pi(s, a) = \Pr\{a = a_t | s = s_t\}$; that decision is sent back to the environment, and the environment gives a reward signal $r(s_t, a_t)$ and transits to the next state $s_{t+1}$ based on the state transition probability $p_{ss'}^a = \Pr\{s' = s_{t+1} | s = s_t, a = a_t\}$. This process repeats until the agent encounters a terminal state after which the process restarts. The main target of reinforcement learning is to maximize the expected discounted return $\mathbb{E}_\pi[\sum_{t=0}^\infty \gamma^t r_t]$ in an unknown environment, where $\gamma \in (0, 1]$ is a factor that balances the importance of future reward. Without information about environment dynamics and task-specific rewards in advance, the agent needs exploration to discover potential valuable states. Apparently, the learned policy may be sub-optimal if the exploration strategy cannot lead to explore the whole state space.

**Intrinsically Motivated Goal Exploration Process.** Intrinsically motivated goal exploration process (IMGEP) (Baranes & Oudeyer, 2009; Forestier et al., 2017) relies on a goal-conditioned (or goal-parameterized) policy $\pi_g$ for unsupervised exploration. It involves the following steps: 1) selecting an intrinsic or interesting state from the experience buffer as the desired goal; 2) exploring with a goal-conditioned policy $\pi_g(s, a, g) = \Pr\{a_t = a | s_t = s, g_t = g\}$; 3) reusing experience for an exploitaion policy $\pi_e(s, a) = \Pr\{a_t = a | s_t = s\}$ to maximize the external reward. Note that the performance of exploitation policy $\pi_e$ relies on samples collected by the goal-exploration policy $\pi_g$. Thus, the criterion of goal selection is crucial for IMGEP.

**Maximum State Entropy Exploration.** Maximum state entropy exploration (Hazan et al., 2019) aims to search an exploration policy $\pi^*$ such that it maximizes the entropy of induced state distribution (or minimizes the KL-divergence between the uniform distribution and induced state distribution) among the class of stationary policies (i.e., $\pi^* \in \arg\max_\pi H[d_\pi]$, where $d_\pi$ is the state distribution induced by $\pi$). Without any information about tasks given by the environment, we think maximum state entropy exploration is safe for exploitation.

## 3 IMGEP WITH MAXIMUM STATE ENTROPY EXPLORATION

In this section, we bridge the intrinsically motivated goal exploration process and maximum state entropy exploration. We begin with practical considerations when maximizing state entropy, then analyze the exploration characteristics of the proposed goal-selection method for IMGEP.

In practice, an exact density estimator for high-dimension state space is intractable, and the state space is unknown, which leads to an empirical state distribution over visited states. The differences are important. For example, directly optimizing the entropy of empirical state distribution over visited states is not what we want, because it ignores the non-visited states outside of the empirical state distribution (see the top row in Fig 2). Instead, we need to first maximize the support of induced state distribution (i.e., discovering new states), then we maximize the entropy of induced state distribution with full support (see the bottom row in Fig 2). In the following, we demonstrate that selecting the states with the least visitation counts among visited states as goals can achieve the above functions under some assumptions.

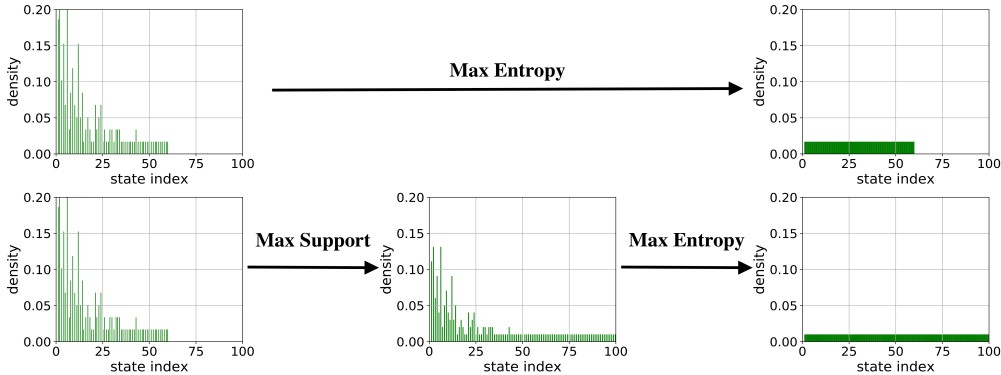

Figure 2: Histograms for normalized state visitation counts, where the $x$-axis represents the index of state. Top row: directly maximizing the entropy of empirical state distribution over visited states; Bottom row: firstly maximizing the counting measure of induced state distribution support, then maximizing the entropy of state distribution with full support.

Let the set $\{1, 2, \cdots, |S|\}$ denotes the state space $S$, $\pi_{1:t}$ denotes the set of policies $\{\pi_1, \pi_2, \cdots, \pi_t\}$ over previous iterations, $\pi_{t+1}$ denotes the policy of next iteration, $x_t^i$ denotes the cumulative visitation counts of state $i$ induced by history policies $\pi_{1:t}$, and $N_t = \sum_{i=1}^{|S|} x_t^i$ denotes the sum of all state visitation counts. Hence, the entropy of empirical state distribution induced by policies $\pi_{1:t}$ is defined as $H[d_{\pi_{1:t}}(s)] = \sum_{i=1}^{|S|} \frac{x_t^i}{N_t} \log \frac{x_t^i}{N_t}$ ($H_t$ for short), and the counting measure of empirical state distribution support induced by policies $\pi_{1:t}$ is defined as $\mu[d_{\pi_{1:t}}(s)] = \sum_{i=1}^{|S|} \mathbb{I}(x_t^i \geq 1)$ ($\mu_t$ for short), where $\mathbb{I}$ is the indicator function.

The theoretical analysis starts with the situation that each iteration the goal-conditioned exploration policy can only select a state to visit without consideration of trajectories towards the goal. Our question is which state to visit gives the most benefits in terms of maximum state entropy. This question is closely related to the goal generation in IMGEP. To facilitate the analysis, let the unit vector $e = [0, \cdots, 1, \ldots] \in \mathbb{R}^{|S|}$ denotes a choice (i.e., $e(i) = 1$ indicates that the policy selects $i$-th state to visit). Note that $x_{t+1} = x_t + e_t$ with this assumption.

**Proposition (Max Counting Measure of Support)** *For any state $i \in \{1, \cdots, |S|\}$ with $x_t^i \geq 0$, unless the unvisited state sets $K = \{i | x_t^i = 0\}$ is an empty set, for any choice $e_t$ such that $e_t(i) = 1$ with $x_t^i = 0$, we have $\mu_{t+1} = \mu_t + 1$.*

This Proposition states visiting the non-visited states is to maximize the counting measure of induced state distribution support. The agent improves its policy by discovering new valuable states. In practical applications, we don't have access to non-visited states in advance. In other words, we can't select these non-visited states as goals since they are not contained in the experience buffer. To deal with this problem, we assume that the chance of discovering non-visited states is high when the agent perform random actions to explore around the exploration boundary. The exploration boundary can be understood as the set of visited states with the least visitation counts (See Figure 1 for the illustration). Our assumption is based on the fact that the total visitations counts of the visited region are large and the total visitation counts of the non-visited region are small. In conclusion, the goal-conditioned exploration policy is asked to reach the exploration boundary, then it performs random actions to discover new states to maximize the counting measure.

**Theorem 1 (Max Entropy)** *For any state* $i \in \{1, \cdots, |S|\}$ *with* $x_t^i \geq 1$; *for any choice* $\boldsymbol{e}_t^*$ *such that* $\boldsymbol{e}_t^*(i) = 1$ *with* $i \in \arg\min_j x_t^j$, *we have* $\boldsymbol{e}_t^* \in \arg\max_{\boldsymbol{e}_t} H_{t+1}$.

We provide the proof in the appendix A.1. Theorem 1 characterizes the behavior of visiting the states with the least visitations when the whole state space has been explored (i.e., the stage after maximizing the counting measure of induced state distribution support). Since Theorem 1 still suggests selecting states with the least visitation counts as goals, the above method can also be applied to maximize the entropy of induced state distribution. Actually, it is easy to unify the two stages via a smoothed entropy $H_\sigma(d_\pi) = -\mathbb{E}_{d_\pi}[\log(d_\pi) + \sigma]$ (Hazan et al., 2019). For our problem, the definition of entropy is proper by assigning non-visited states with a "dummy" visitation counts between 0 and 1. In that case, Theorem 1 still holds and suggests firstly selecting these non-visited states and subsequently selecting the states with least visitation counts to maximize the smoothed state entropy.

The proposed exploration method is called **novelty-pursuit**. We notice that the above analysis neglects the influences of trajectories towards the exploration boundary. However, the fluctuation of state distribution entropy by the trajectories towards the exploration boundary is less significant from practical considerations. In fact, the goal-conditioned policy should be trained to reach the exploration boundary quickly and pays more efforts to discover new states around the exploration boundary, as our experiment results in Section 5.1 indicate.

# 4 METHOD

In this section, we present practical implementations for the proposed method. How to approximate visitation counts in high-dimension space and how to estimate the exploration boundary is given in Section 4.1. We describe the training technique of goal-conditioned policy in Section 4.2. Finally, we introduce an exploitation policy to learn the experience collected by the goal-conditioned exploration policy in Section 4.3. We outline the proposed exploration method in Algorithm 1.

## 4.1 APPROXIMATING EXPLORATION BOUNDARY IN HIGH-DIMENSION SPACE

Generally, computing the visitation counts in high-dimension space is intractable. However, it is possible to build some variables related to the visitation counts. For example, Burda et al. (2019b) show that prediction errors given by two randomly initialized network have a strong relationship to the number of training samples on the MNIST dataset. Thus, we can use the prediction errors to sort visited states. Other approaches like pseudo-counts (Bellemare et al., 2016; Ostrovski et al., 2017) can be also applied, but we find that RND is easy to scale up.

RND is consist of two randomly initialized neural networks: a fixed network called target network $f(x; \omega_t)$, and a trainable network called predictor network $\hat{f}(x; \omega_p)$. Both two networks take a state $s$ as input and output a vector with the same dimension. Each time a batch of data feeds into the predictor network to minimize the difference between the predictor network and the target network concerning the predictor network's parameters, shown in Equation 1.

$$\min_{\omega_p} \frac{1}{K} \sum_{i=1}^{K} ||f(s_i; \omega_t) - \hat{f}(s_i; \omega_p)||^2 \qquad (1)$$

In practice, we employ an online learning setting to train RND and maintain a priority queue to store states with the highest prediction errors. In particular, after a goal-conditioned policy collects a mini-batch of transitions, this data feed to train the predictor network. Also, a state with high prediction error will be stored into the priority queue and the state with the least prediction error will be removed out of the priority queue if full. This process repeats and no historical data will be reused to train the predictor network. Besides, each iteration a state will be selected from the priority queue as a goal for the goal-conditioned policy. After achieving the goal, the exploration policy will perform random actions to discover new states. Consider the bias due to approximation, we sample goals from a distribution based on their prediction errors (e.g., softmax distribution).

---

**Algorithm 1** Exploration by novelty-pursuit

---

**Input:** predictor network update interval $K$; goal-conditioned policy update interval $M$; mini-batch size of samples for goal-conditioned policy $N$;
Initialize parameter $\theta$ for goal-conditioned exploration policy $\pi_g(s, g, a; \theta)$.

Initialize parameter $\omega_t$ for target network $f(x; \omega_t)$, and $\omega_p$ for predictor network $\hat{f}(x; \omega_p)$.
Initialize a buffer $D_g$ for $\pi_g$, and a priority queue $Q$ to store states with least visitation counts.
**for** each iteration **do**
    Reset the environment and get the observation $o_0$;
    Choose a goal $g$ from priority queue $Q$, and set $goal\_success = False$;
    **for** each timestep $t$ **do**
        **if** $goal\_success == True$ **then**
            Choose an random action $a_t$; **# Explore around the exploration boundary**
        **else**
            Choose an action $a_t$ from $\pi_g(s_t, g, a_t; \theta)$; **# Go to the exploration boundary**
        **end if**
        Send $a_t$ to the environment and get $r_t^e, s_{t+1}$;
        Update $goal\_success(s_{t+1}, g)$;
        **# Store new states and update the predictor network**
        **if** $t \% K == 0$ **then**
            Store transitions $\{s_k, g, a_k, r_k^e\}_{k=t-K}^t$ into replay buffer $D_g$;
            Calculate prediction errors for $\{s_k\}_{k=t-K}^t$ and store them into priority queue $Q$;
            Update predictor network $\hat{f}(x; \omega_p)$ using $\{s_k\}_{k=t-K}^t$;
        **end if**
        **# Update $\pi_g$ with reward shaping**
        **if** $t \% M == 0$ **then**
            Update $\pi_g$ with $\{s_k, g_k, a_k, r_k^i\}_{k=1}^K$ sampled from $D_g$;
        **end if**
    **end for**
**end for**

---

## 4.2 Training goal-conditioned policy efficiently

Before we describe the training techniques for the goal-conditioned policy, we emphasize that training this policy doesn't require the external reward signal from the environment. But we additionally use the external reward for the goal-conditioned policy to reduce the mismatch behaviors between the goal-conditioned policy $\pi_g$ and the exploitation policy $\pi_e$.

Following multi-goal reinforcement learning (Andrychowicz et al., 2017; Plappert et al., 2018a), we manually extract goal information from state space. Specifically, each state $s$ is associated with an achieved goal of $ag$, and the desired goal is denoted as $g$. To avoid ambiguity, a goal-conditioned policy $\pi_g(s, a, g; \theta)$[1] is asked to accomplish a desired goal $g$. For our settings, the achieved goal is coordinate information.

$$r(ag_t, g_t) = \begin{cases} 1 & \text{if } d(ag_t, g_t) < \epsilon \\ 0 & \text{otherwise} \end{cases} \tag{2}$$

A proper reward function for the goal-conditioned policy is an indicator function with some tolerance, shown in Equation 2. With a little abuse of notations, between the achieved goal $ag$ and the desired goal $g$ we use $d(ag, g)$ to denote some "distance" (e.g., $L_1$ or $L_2$ norm) between the achieved goal $ag$ and the desired goal $g$. If the distance is less than some threshold $\epsilon$, the goal-conditioned policy receives a positive reward otherwise zero. Note that this function is also be used to judge whether agents reach the exploration boundary. However, the training of goal-conditioned policy is slow with this sparse reward function. Next, we introduce some techniques to deal with this problem.

$$r(ag_t, g_t) = d(ag_{t-1}, g_t) - d(ag_t, g_t) \tag{3}$$

Reward shaping introduces additional training rewards to guide the agent. Reward shaping is invariant to the optimal policy if shaping reward function is a potential function (Ng et al., 1999).

---

[1]With the respect of input to a goal-conditioned policy, $s$ contains $ag$ to keep notations simple.

Specifically, we define the difference of two consecutive distances (between the achieved goal and the desired goal) as shaping reward function, shown in Equation 3. Since shaping reward function is dense, it can lead to substantial reductions in learning time. Verification of the optimal goal-conditioned policy is invariant between this function and the indicator reward function is given in Appendix A.2. Alternatively, one can use also Hindsight Experience Replay (HER) (Andrychowicz et al., 2017) to train the goal-conditioned policy via replacing each episode with an achieved goal rather than one that the agent was trying to achieve. But one should be careful since HER changes the goal distribution for learning. Besides, one can also utilize past trajectories to accelerate training, which we discuss in Appendix A.3.

### 4.3 EXPLOITING EXPERIENCE FROM EXPLORATION POLICY

Parallel to the goal-conditioned exploration, we additionally train an unconditioned exploitation policy $\pi_e$, which only takes the state as input. This policy learns from experience collected by the exploration policy $\pi_g$ in an off-policy learning fashion. At the same time, the exploitation policy also interacts with the environment to mitigate the side effect of exploration error (Fujimoto et al., 2019), a phenomenon that off-policy learning degenerates when data from the exploration policy is not correlated to the experience generated by the exploitation policy. Note that exploitation policy is trained with an RL objective to maximize expected discounted external return. Therefore, the exploration and exploitation are naturally decoupled, which turns out to help escape the local optimum on SuperMarioBros environments. From this perspective, our method is distinguished from Go-Explore Ecoffet et al. (2019), which employs exploration followed by exploitation.

## 5 EXPERIMENT

In this section, we aim to answer the following research questions: 1) Does novelty-pursuit effectively maximize the state entropy? 2) Do the proposed goal-selection criterion and training techniques improve performance for IMGEP? 3) How does the performance of novelty-pursuit compare with the state-of-the-art approaches in complex environments? We conduct experiments from the simple maze environments, Mujoco tasks, to long-horizon video games of SuperMarioBros to evaluate the proposed method. Detailed policy network architecture and hyperparameters are given in Appendix A.6 and A.7, respectively.

Here we briefly describe the environment settings (see Figure 3 for illustrations). Detailed settings are given in the Appendix A.5.

**Empty Room & Four Rooms.** An agent navigates in the maze of $17 \times 17$ to find the exit (Chevalier-Boisvert et al., 2018). The agent receives a time penalty until it finds the exit and receives a positive reward. The maximum return for both two environments is $+1$, and the minimum total reward is $-1$. Note that the observation is a partial image of shape $(7, 7, 3)$.

**FetchReach.** A 7-DOF Fetch Robotics arm (simulated in the Mujoco (Todorov et al., 2012)) is asked to grip spheres above a table. There are 4 spheres on the table, and the robot receives a positive reward of $+1$ when its gripper catches a sphere (the sphere will disappear after being caught) otherwise it receives a time penalty. The maximum total reward is $+4$, and the minimum total reward is $-1$.

**SuperMarioBros.** A Mario agent with raw image observation explores to discover the flag. The reward is based on the score given by the NES simulator (Kauten, 2018) and is clipped into $-1$ and $+1$ except $+50$ when getting a flag. There are 24 stages in the game, but we only focus on the 1-1, 1-2, and 1-3.

### 5.1 COMPARISON OF EXPLORATION EFFICIENCY

In this section, we study the exploration efficiency in terms of the state distribution entropy. We focus on the Empty Room environment because it is tractable to calculate the state distribution entropy. Note that we don't use any external reward the observation for RND is a local-view image.

We consider the following baselines: 1) random: uniformly selecting actions; 2) bonus: a policy receiving exploration bonus based on the prediction errors of RND (Burda et al., 2019b); 3) novelty-

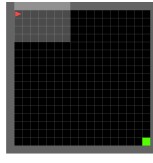 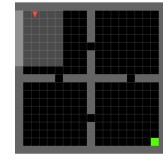 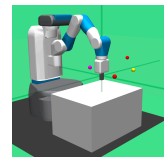 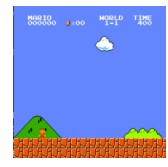

(a) Empty Room  (b) Four Rooms  (c) FetchReach  (d) SuperMarioBros

Figure 3: Illustration of four environments considered in this paper.

pursuit: the proposed method. We also consider three variants of our method: 4) novelty-pursuit-planning oracle: the proposed method with a perfect goal-conditioned policy; 5) novelty-pursuit-counts-oracle: the proposed method with selecting goals based on true visitation counts; 6) novelty-pursuit-oracles: the proposed method with both two oracles. The results are summarized in Table 1. Note that the maximum state distribution entropy for this environment is 5.666.

Table 1: Average entropy of visited state distribution at timesteps $200k$ over 5 seeds on Empty Room.

|  | Entropy |
| --- | --- |
| random | $5.129 \pm 0.021$ |
| bonus | $5.138 \pm 0.085$ |
| novelty-pursuit | $\mathbf{5.285 \pm 0.073}$ |
| novelty-pursuit-planning-oracle | $5.513 \pm 0.077$ |
| novelty-pursuit-counts-oracle | $5.409 \pm 0.059$ |
| novelty-pursuit-oracles | $5.627 \pm 0.001$ |
| maximum | 5.666 |

First, we can see that novelty-pursuit achieves a higher entropy than the random and bonus method. Though exploration bonus via prediction errors of RND may help makes an exploration-exploitation trade-off (Burda et al., 2019b), but is inefficient to a maximum state entropy exploration. We attribute this to delayed and indirect feedbacks of the exploration bonus. Second, when the planning oracle and visitation counts oracle are available, the entropy of our method roughly improves by $0.228$ and $0.124$, respectively. We notice that the planning-oracle avoids exploration towards the exploration boundary and spends more meaningful steps to explore around the exploration boundary, thus greatly improves the entropy. Based on this observation, we think accelerating goal-conditioned policy training is more important for our method. Actually, we find the proposed method can satisfy our need to approximate the exploration boundary via prediction errors of RND (See Appendix A.4 for more results). Third, the combination of two oracles gives a near-perfect performance (the gap between the maximum state entropy is only $0.039$). This result demonstrates that goal-condition exploration behaviors presented by novelty-pursuit can maximize the state entropy and validates the analysis in Section 3.

### 5.2 ABLATION STUDY OF GOAL-SELECTION AND TRAINING TECHNIQUES

In this section, we study the factors that contribute to our method by ablation experiments. Firstly, we focus on the criterion of goal-section in IMGEP. We compare novelty-pursuit with two other goal-selection methods: 1) random-selection: selecting states randomly from the experience buffer; 2) learning-progress: selecting a feasible state (goal success rate is between $0.3$ and $0.7$) with probability of $0.8$ and an arbitrary visited state with the probability of $0.2$, which is adopted from (Forestier et al., 2017). Results on the Empty Room are shown in Figure 4. Secondly, we study how goal-conditioned policy learning affects performance. We compare HER and the reward-shaping with distance reward (i.e., reward based on $L_1$ norm in our problem) used in (Forestier et al., 2017). Results on the Empty Room are shown in Figure 5.

From Figure 4, we see that IMGEP doesn't work when randomly selecting goals, but novelty-pursuit gives a greater boost compared to the learning-progress. We think the reason is that this heuristic method is brittle to the estimation of goal success rate and lacks an explicit exploration objective.

From Figure 5, we find that the IMGEP with HER or reward shaping outperforms than the IMGEP with distance reward. As discussed in Ng et al. (1999), reward based on distance may change the optimal behavior of goal-condition exploration policy, thus hurts the performance for IMGEP.

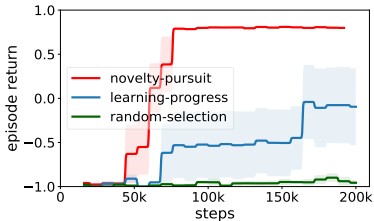
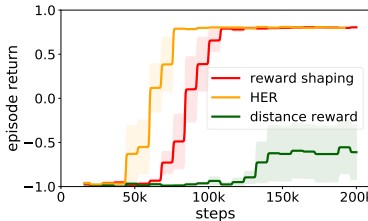

Figure 4: Comparison of goal-selection.     Figure 5: Comparison of training techniques.

### 5.3 EVALUATION ON COMPLEX ENVIRONMENTS

In this section, we compare different methods in terms of external reward. We will see that without sufficient and efficient exploration, the policy may be stuck into the local optimum. Two baseline methods using reinforcement learning are considered: 1) vanilla: DDPG (Lillicrap et al., 2016) with Gaussian action noise on Fetch Reach and ACER (Wang et al., 2017) with policy entropy regularization on others; 2) bonus: an off-policy version of (Burda et al., 2019b) that combines the external reward and intrinsic reward based on the vanilla policy. Note reported results of novelty-pursuit are the performances of the exploitation policy $\pi_e$ rather than the goal-conditioned exploration policy $\pi_g$. We keep the number of samples and training iterations same for all methods.

First, we consider the previously used Empty Room and the Four Room environments. The results are shown in Figure 6. We see that the vanilla policy hardly finds the exit. Novelty-pursuit is comparative to bonus and outperforms bonus on the Four Rooms environment, where we observe that bonus is somewhat misled by the intrinsic reward though we have tried many weights to balance the external reward and intrinsic reward.

Secondly, we consider the FetchReach environment and results are shown in Figure 6. We see that novelty-pursuit can consistently grip 4 spheres while other methods sometimes fail to efficiently explore the whole state space to grip 4 spheres.

Finally, we consider the SuperMarioBros environments, in which it is very hard to discover the flag due to the huge space state and the long horizon. Learning curves are plotted in Figure 7 and the final performance is listed in Table 2. We find the vanilla method gets stuck into the local optimum on SuperMarioBros-1-1 while the bonus method and ours can find a near-optimal policy. All methods perform well on SuperMarioBros-1-2 thanks to dense rewards. On SuperMarioBros-1-3, reward is sparse and the task is very challenging. We plot trajectories of SuperMarioBros-1-3 in Figure 8, and

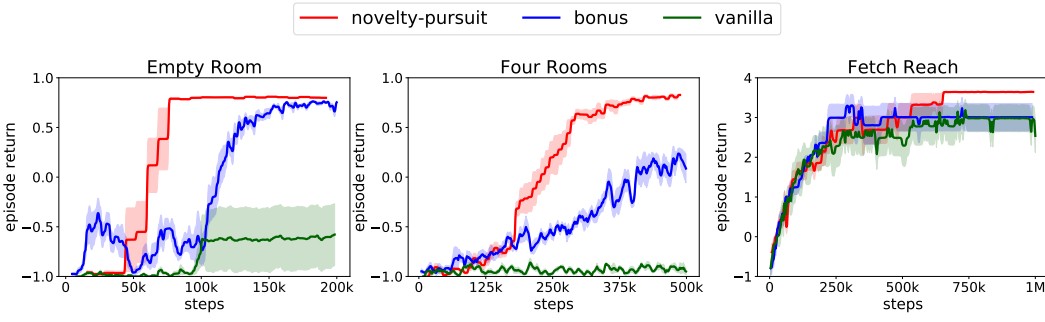

Figure 6: Average episode returns over 5 seeds on the Empty Room, Four Rooms and FetchReach environments. Shadows indicate the standard deviation.

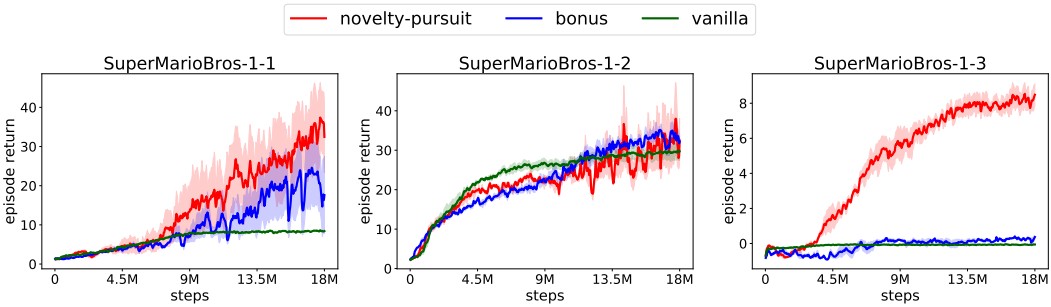

Figure 7: Average episode returns over 3 seeds on SuperMarioBros. Shadows indicate the standard deviation.

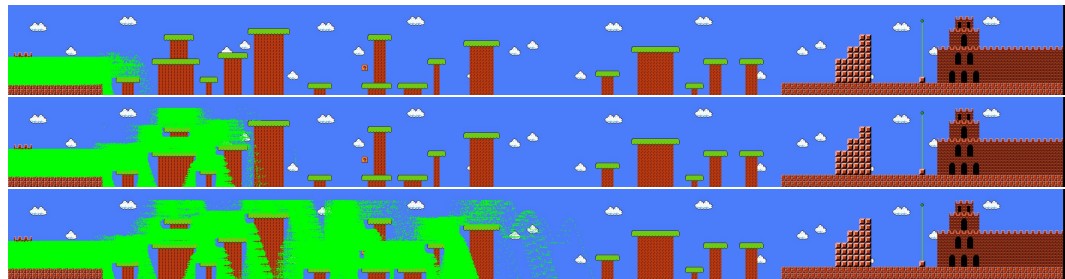

Figure 8: Trajectory visualization on SuperMarioBros-1-3. Trajectories are plotted in green cycles with the same number samples (18M). The agent starts from the most left part and needs to fetch the flag on the most right part. Top row: vanilla; middle row: bonus; bottom row: novelty-pursuit.

more results can be found in Appendix A.4. It turns out only our method can get positive rewards via a deep exploration presented by the goal-conditioned policy on SuperMarioBros-1-3.

Table 2: Final Performance over 3 seeds on SuperMarioBros.

|  | novelty-pursuit | bonus | vanilla |
|---|---|---|---|
| SuperMarioBros-1-1 | **36.02 ± 8.19** | 17.74 ± 7.84 | 8.43 ± 0.14 |
| SuperMarioBros-1-2 | **33.30 ± 6.13** | **33.19 ± 1.53** | 29.64 ± 2.02 |
| SuperMarioBros-1-3 | **8.14 ± 0.55** | 0.20 ± 0.14 | -0.07 ± 0.01 |

## 6 RELATED WORK

**Exploration.** Traditionally, the exploration strategy is based on the exploitation policy that receives an external reward from the environment. Traditional exploration methods include injecting noise on action space (Mnih et al., 2015; Lillicrap et al., 2016) or parameter space (Plappert et al., 2018b; Fortunato et al., 2018), and adding the policy's entropy regularization (Schulman et al., 2017; Mnih et al., 2016).

For tabular Markov Decision Process, there are lots of work utilizing confidence based reward to balance exploration and exploitation (Kearns & Singh, 2002; Strehl & Littman, 2008; Kolter & Ng, 2009; Lattimore & Hutter, 2014). Several exploration strategies for deep RL based approximation visitation counts have been proposed in high-dimension space (Bellemare et al., 2016; Ostrovski et al., 2017). Another type of exploration is curiosity-driven exploration. These methods track the uncertainty of dynamic (Stadie et al., 2015; Pathak et al., 2017; Burda et al., 2019a;b) to explore intrinsic states. Deep (temporally extended) exploration via tracking the uncertainty of value function is studied in (Osband et al., 2016). Besides, maximum (policy) entropy reinforcement learning

encourages exploration by maximizing the cumulative sum of external reward and policy entropy (Ziebart et al., 2008; Haarnoja et al., 2017; O'Donoghue et al., 2016; Haarnoja et al., 2018).

Recently, Hazan et al. (2019) introduce a new exploration objective: maximum state entropy. They provide an efficient algorithm when restricted to a known tabular MDP (a density estimator oracle is required for an unknown tabular MDP) and gives the theoretical analysis. We derive the criterion of goal generation based on the principle of maximum state entropy.

Our method is based on the framework of intrinsically motivated goal exploration processes (IMGEP) (Baranes & Oudeyer, 2009; Forestier et al., 2017; Péré et al., 2018). Go-Explore (Ecoffet et al., 2019) is reminiscent of IMGEP and achieves dramatic improvement on the hard exploration problem of Montezumas Revenge. But with the assumption that the environments are resettable or deterministic and many hand-engineering designs, Go-Explore is restricted to specific environments. Our method shares a similar exploration strategy like Go-Explore, but our method is implemented practically and can be applied to stochastic environments. Importantly, we aim to answer the core question: why such defined goal-conditioned exploration is efficient?

**Goal-conditioned Policy.** By taking environment observation and desired goal as inputs, the goal-conditioned policy is expected to accomplish a series of tasks. Schaul et al. (2015) propose the universal value function approximator (UVFA) and train it by bootstrapping from the Bellman equation. However, training goal-condtioned policy is also still a challenging problem due to goal-condition reward is sparse (e.g. 1 for success, 0 for failure). Andrychowicz et al. (2017) propose hindsight experience replay (HER) by replacing each episode with an achieved goal rather than one that the agent was trying to achieve. This operation introduces more reward signals and serves as an implicit curriculum. Florensa et al. (2018) use a generator network to adaptively produce artificial feasible goals. We also use a goal-conditioned policy, but goals are selected from the experience buffer rather than being specified in advance. What's more, we utilize the technique of reward shaping (Ng et al., 1999) to accelerate training.

**Learning from experience.** Off-policy reinforcement learning algorithms such as DQN(Mnih et al., 2015), DDPG (Lillicrap et al., 2016), and ACER (Wang et al., 2017), reuse experience to improve data efficiency. Besides, how to additionally utilize (good) experience to overcome exploration dilemma is studied in (Oh et al., 2018; Goyal et al., 2019). These works are perpendicular to ours since we focus on how to discover these valuable states.

## 7 CONCLUSION

This paper bridges the intrinsically motivated goal exploration process (IMGEP) and the maximum state entropy exploration (MSEE). We propose a method called novelty-pursuit from the connection. We demonstrate the proposed method is efficient towards exploring the whole state space. Therefore, the proposed method can escape from the local optimum and heads the (near-) optimal policy. We notice that current training techniques of the exploitation policy are based on an RL objective, which may not be efficient to utilize experience collected by the exploration policy. Theoretically, the influence of trajectories towards the exploration bound should be considered. We leave these for future works.

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

# A    APPENDIX

## A.1    PROOF OF THEOREM 1

Suppose we have two choices among state $i$ and state $j$, we want to compare the difference $g(i,j)$ between the entropy $H_i[d_{\pi_{1:t+1}}]$ by visiting state $i$ and the entropy $H_j[d_{\pi_{1:t+1}}]$ by visiting state $j$. Let $N_t = \sum_i x_t^i$ denotes visitation counts over all states. Note that the entropy difference of two choices can be attributed to changes in $x_{t+1}^i$ and $x_{t+1}^j$:

$$
\begin{aligned}
g(i,j) &= H_i[d_{\pi_{1:t+1}}] - H_j[d_{\pi_{1:t+1}}] \\
&= \left(-\frac{x_t^i+1}{N_t+1}\log\frac{x_t^i+1}{N_t+1} - \frac{x_t^j}{N_t+1}\log\frac{x_t^j}{N_t+1}\right) - \left(-\frac{x_t^j+1}{N_t+1}\log\frac{x_t^j+1}{N_t+1} - \frac{x_t^i}{N_t+1}\log\frac{x_t^i}{N_t+1}\right) \quad (4) \\
&= \left(\frac{x_t^j+1}{N_t+1}\log\frac{x_t^j+1}{N_t+1} - \frac{x_t^j}{N_t+1}\log\frac{x_t^j}{N_t+1}\right) - \left(\frac{x_t^i+1}{N_t+1}\log\frac{x_t^i+1}{N_t+1} - \frac{x_t^i}{N_t+1}\log\frac{x_t^i}{N_t+1}\right)
\end{aligned}
$$

Let $f(x) = \frac{x+1}{N_t+1}\log\frac{x+1}{N_t+1} - \frac{x}{N_t+1}\log\frac{x}{N_t+1}$, which yields

$$g(i,j) = f(x_t^j) - f(x_t^i) \qquad (5)$$

By looking at the derivative of $f(x)$, we know that $f(x)$ is a monotonically increasing function. Thus, for any $x_t^i < x_t^j$, we have that $g(i,j) > 0$.

$$f'(x) = \frac{1}{N_t+1}\log(1+\frac{1}{x}) > 0 \qquad (6)$$

In conclusion, unless state $i$ has the least visitation counts, we can always another state $j$ with $x_t^j < x_t^i$ to increase the entropy. Hence, visiting the states with the smallest visitation counts is optimal.

## A.2    REWARD SHAPING FOR MULTI-GOAL REINFORCEMENT POLICY

Reward shaping is invariant to the optimal policy under some conditions (Ng et al., 1999). Here we verify that reward shaping introduced by our method doesn't change the optimal policy for goal-conditioned policy. Adding up shaping rewards gives:

$$
\begin{aligned}
\sum_{t=1}^{T} &- d(ag_t, g) + d(ag_{t+1}, g) \\
&= -d(ag_1, g) + d(ag_2, g) - d(ag_2, g) + d(ag_3, g) + \cdots - d(ag_T, g) + d(ag_{T+1}, g) \\
&= -d(ag_1, g) + d(ag_{T+1}, g)
\end{aligned}
\qquad (7)
$$

For the optimal policy $\pi_g$, $d(ag_{T+1}, g) = 0$, while $d(ag_1, g)$ is a constant. Thus, for the fixed $g$, the optimal policy $\pi_g$ induced by the reward shaping is invariant to the one induced by sparse-reward in Equation 2.

### A.3 TRAINING GOAL-CONDITIONED POLICY WITH PAST TRAJECTORIES

In fact, training the goal-conditioned policy for our problem is different from the settings of multi-goal reinforcement learning (Andrychowicz et al., 2017; Plappert et al., 2018a). The goal is selected from visited states rather than the non-visited states. Thus, we can utilize past trajectories to accelerate training with supervised learning. The optimization problem is defined in Equation 8. Note that we cannot rely on this information on stochastic environments like SuperMarioBros.

$$\min_\theta \sum_{(a_t, o_t) \sim \tau(g)} -\log \pi_g(a_t | o_t, g; \theta) \qquad (8)$$

where $\tau(g)$ is the trajectory that covers the goal $g$ in the previous exploration process.

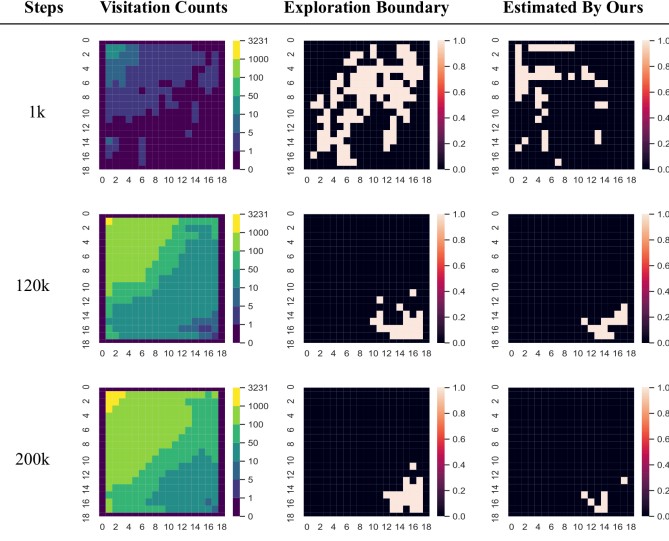

Figure 9: Visualization for the exploration boundary given by visitation counts and the estimated via prediction errors of RND.

### A.4 ADDITIONAL RESULTS

**Empty Room.** We depict the exploration boundary by visitation counts and the estimated one by our method in Figure 9. The agent starts from the left top corner and performs a random policy. The exploration boundary shown in black is top 10% states with least visitation counts or the largest prediction errors among all visited states.

**SuperMarioBros.** In Figure 10, we make additional trajectories visualization on SuperMarioBros-1-1 and SuperMarioBros-1-2. Trajectories are plotted with same number samples (18M). Vanilla method gets into the local optimum even with policy entropy regularization on SuperMarioBros-1-1. In addition, only our method can get the flag on SuperMarioBros-1-2.

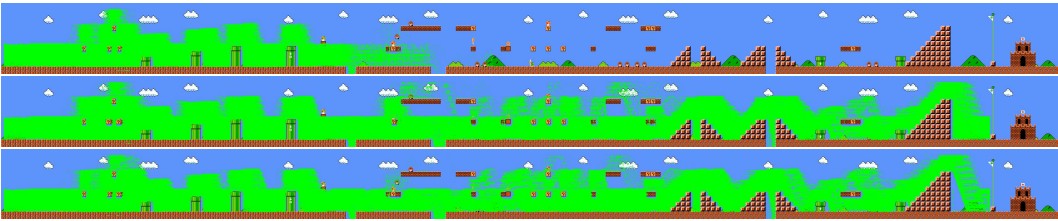

(a) SuperMarioBros-1-1. Agent starts from the most left part and needs to find the flag on the most right part.

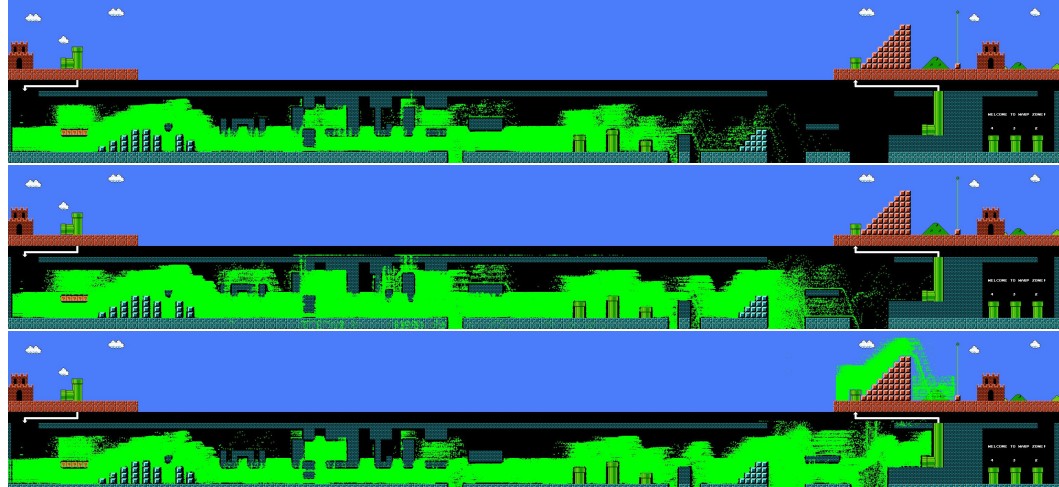

(b) SuperMarioBros-1-2. The agent walks in the underworld shown in black and needs to get the flag through the water pipe on the right part (see arrows).

Figure 10: Trajectory visualization. For each figure, top row: vanilla (ACER); middle row: bonus; bottom role: novelty-pursuit (ours). The vanilla method gets stuck into the local optimum even with policy entropy regularization on SuperMarioBros-1-1. Only our method can get the flag on SuperMarioBros-1-2.

### A.5 ENVIRONMENT PREPOSSESSING

**Maze.** Different from (Chevalier-Boisvert et al., 2018), we only use the image and coordination information as inputs. We only consider four actions: turn left, turn right, move forward and move backward. The maximum episode length is 190 for Empty Room, and 500 for Four Rooms. Each time the agent receives a time penalty of $1/\mathrm{max\_episode\_length}$ and receives +1 when finding the exit.

**FetchReach.** We implement this environment based on *FetchReach-v0* in Gym (Brockman et al., 2016). The maximum episode length is 50. The locations of four spheres are (1.20, 0.90, 0.65), (1.10, 0.72, 0.45), (1.20, 0.50, 0.60), and (1.45, 0.50, 0.55). When sampling goals, we remove spheres outside of the table i.e., the valid x range: (1.0, 1.5), the valid y range is (0.45, 1.05), and valid z range is (0.45, 0.65).

**SuperMarioBros.** We implement this environment based on (Kauten, 2018) with Gym wrappers. Prepossessing includes grey-scaling, observation downsampling, external reward clipping (except that 50 for getting flag), stacked frames of 4, and sticky actions with a probability of 0.25. The maximum episode length is 800. The environment restarts when the agent dies.

### A.6 NETWORK ARCHITECTURE

We use the convolutional neural network (CNN) for Empty Room, Four Rooms, and video games of SuperMarioBros, and multi-layer perceptron (MLP) for FetchReach environment. Network architecture design and parameters are based on baselines (Dhariwal et al., 2017). For each environment,

RND uses a similar network architecture. The predictor network has additional MLP layers than the predictor network.

## A.7 HYPERPARAMETERS

Table 3 gives hyperparameters for ACER (Wang et al., 2017) on the maze and SuperMarioBros (the learning algorithm is RMSProp (Tieleman & Hinton, 2012)). DDPG (Lillicrap et al., 2016) used in Fetch Reach environments is based on the HER algorithm implemented in baselines (Dhariwal et al., 2017) expect that the actor learning rate is 0.0005. We run 4 parallel environments for DDPG and the size of the priority queue is also 100. As for the predictor network, the learning rate of the predictor network is 0.0005 and the optimization algorithm is Adam (Kingma & Ba, 2015) for all experiments, and the batch size of training data is equal to the product of rollout length and the number of parallel environments.

The goal-conditioned policy is trained with shaping rewards defined in Equation 3 and external rewards, which helps reduce mismatch behaviors between its and the exploitation policy's . The weight is 1 for all environments except 2 for SuperMarioBros. For bonus method used in Section 5, the weight $\beta$ to balance the exploration bonus and the external reward (i.e., $r' = r_{ext} + \beta r_{int}$) is 0.1 for Empty Room and Four Rooms, 0.01 for FetchReach, 1.0 for SuperMarioBros-1-1 and SuperMarioBros-1-3, and 0.1 for SuperMarioBros-1-2. We also do a normalization for the intrinsic reward by dividing the intrinsic rewards via a running estimate of the standard deviation of the sum of discounted intrinsic rewards.

Table 3: Hyperparameters of our method based on ACER on the maze and SuperMarioBros.

| Hyperparameters | Empty Room | Four Rooms | SuperMarioBros |
|---|---|---|---|
| Rollout length | 20 | 20 | 20 |
| Number of parallel environments | 4 | 4 | 8 |
| Learning rate | 0.0007 | 0.0007 | 0.00025 |
| Learning rate schedule | linear | linear | constant |
| $\gamma$ | 0.95 | 0.95 | 0.95 |
| Entropy coefficient | 0.10 | 0.10 | 0.10 |
| Size of priority queue | 100 | 100 | 20 |
| Total training steps | 200K | 500K | 20M |

