# OpenReview forum: "Deep exploration by novelty-pursuit with maximum state entropy"
_ICLR.cc/2020/Conference — Reject_

### Official Review · AnonReviewer2 · 2019-10-22
**Official Blind Review #2**

**Rating:** 1

**Review:**

*Summary*
The paper addresses the challenge of intrinsically-driven exploration in tasks with sparse or delayed rewards. First, the authors try to bridge the gap between the objectives of intrinsically-motivated goal generation and maximum state entropy exploration. Then, they propose a new exploration method, called novelty-pursuit, that prescribes the following receipt: first, reach the exploration boundary through a goal-conditioned policy, then take random actions to explore novel states. Finally, the authors compare their approach to a curiosity-driven method based on Random Network Distillation in a wide range of experiments: from toy domains to continuous control, to hard-exploration video games.

I think that the paper displays some appealing empirical and methodological contributions, but it is not sufficiently theoretically grounded. For this reason, I would vote for rejection. I would advise the authors to rephrase their work as a primarily empirical contribution, in order to emphasize the merits of their method over a lacking theoretical analysis.

*Detailed Comments*

*Major Concern*
My major concern is about the claim that goal-conditioned exploration towards the least visited state would, at the same time, maximize the entropy of the state distribution. The derivations seem technically sound, but I think that the underlying assumption is unreasonable in this context: it neglects the influence of the trajectory to reach the target state, which is rather crucial in reinforcement learning instead. It is quite easy to design a counter-example in which the (optimal) goal-conditioned policy towards the least visited state actually decreases the overall entropy of the state distribution. One could avoid the issue by assuming to have access to a generative model over the states, but that would fairly limit the applicability of the approach.

*Other Concerns and Typos*
- I think that the authors minimize the relation between their methodology and the one proposed in (Ecoffet et al., 2019). It is true that the applicability of Go-Explore is quite limited. However, the idea behind their approach, which is based on first reaching an already visited state and then exploring randomly from that state, is not all dissimilar from the two-phase exploration scheme of novelty-pursuit.
- It is not completely clear to me how the disentanglement between exploration and exploitation works in the novelty-pursuit algorithm.
- What is the vanilla policy considered in the experiments?
- Section 4.2, after equation 3: rewarding shaping -> reward shaping
- section 5.4: we consider the SuperMarioBros environments, which is very hard ecc. -> we consider the SuperMarioBros environments, in which it is very hard ecc.


**Experience Assessment:**

I have read many papers in this area.

**Review Assessment: Checking Correctness Of Derivations And Theory:**

I carefully checked the derivations and theory.

**Review Assessment: Checking Correctness Of Experiments:**

I assessed the sensibility of the experiments.

**Review Assessment: Thoroughness In Paper Reading:**

I read the paper at least twice and used my best judgement in assessing the paper.

---

> ### Author Response · Authors · 2019-11-13
> **Replay to Review 2**
>
> Thank you for your helpful review.
> Thanks for your thoughtful suggestions about the improvement of theoretical analysis. We do notice that our analysis ignores the influence of trajectories toward the goals. But we think that, in practice, the fluctuations of entropy introduced by a perfect goal-conditioned policy are less important. The increased part of exploration around the exploration boundary matters. We verify this conjecture in Section 5.1. But we will attempt to improve our theoretical analysis to consider the goal-conditioned trajectories in the future works.
>
> Exploration behaviors of our method are similar to the Go-Explore’s. But we present practical methods. What’ more, the trade-off of exploration and exploitation is different from Go-Explore’s, which we discuss in Section 4.3. Importantly, we attempt to answer the question: why such defined exploration is efficient.
>
> Vanilla policy: DDPG (with Gaussian action noise) on Fetch Reach and ACER (with policy entropy regularization) for others.
>
> We have revised our writing styles according to your suggestions. We appreciate it if you can reconsider after reading the revised paper and the above responses.

---

### Official Review · AnonReviewer1 · 2019-10-23
**Official Blind Review #1**

**Rating:** 3

**Review:**

The authors study the problem of exploration in deep reinforcement learning. The authors borrow the ideas developed in the intrinsically motivated goal exploration process, and entropy maximization and propose a method on Noverly-Pursuit. The authors then empirically study the proposed method.

1) The authors investigate an important problem and I would appreciate the authors if they could motivate its importance more in their work.

2) In the second paragraph, the author mentioned that goal-conditioned exploration behaviours can maximize entropy. Later in the same paragraph, they claim that "The exploration policy leads
to maximize the state entropy on the whole state distribution considering tabular MDP". I guess the authors' point was this approach might increase the entropy rather than maximizing it. If the claim is, in fact, maximization, a reference would be helpful. If the authors prove it in this paper, implying it in this paragraph is also helpful.

3) In the background section, the authors did not specify whether they provide background on tabular MDP or beyond that. By calling the transition kernel the state transition probabilities, it seems they introduced a tabular MDP, but a more concrete introduction and preliminaries would help to follow the paper.

3) The first paragraph of section 2, the author mentioned that
"The target of reinforcement learning is to maximize the expected discounted return". I hope the authors mean"one of the targets in the study of reinforcement learning ..."

4) in the same paragraph, why the \gamma = 0 is excluded? is there any specific reason? Also, when the authors include the gamma = 1, do they make sure the maximization in line 9 of the same paragraph is well defined in regular cases?

5) Regarding the experiment in figure 2. It would be useful to the readers if the authors provide more details about this experimental study.

6) In the few paragraphs below Figure 2, it would be nicer if the authors provide a clear definition of each term. In order to follow the paper, I relied on my imperfect inference to infer the definitions. Also, I find it probably useful to distinguish the random variables and their realizations in the notation.

6) Regarding the theorem1. I would recommend making the statement more transparent and more clear. I also recommend to even not calling it a theorem since it, as mentioned, is as clear as the definitions. Also, arent x_t(i)s non-negative by definition?

7) In this sentence:
"However, we don’t know what non-visited states are and
where non-visited states locate in practice since we can’t access ..."
I think the authors' point was that "we might not have access to it in general".

8) It would be helpful to me to evaluate this paper if the authors explain more how the following statements go through:
"To deal with this problem, we assume that state density over the whole state space is continuous, thus visited states and non-visited states are close". I am not sure how "thus visited states and non-visited states are close" follows from continuity of density and what is the notion of closeness.

9-1) Theorem 2: while H seems to be a function of d_\pi1:t(s), I am not sure how to interpret the argmax_{e_t}. A bit of help from the authors would be appreciated.

9-2) Theorem 2: In the proof, I was not able to justify to my self the transition form g(xi; xj) = Hxi [d1:t+1] 􀀀 Hxj [d1:t+1] to the second line. Also, what is the definition of H_x_i?

10) In equation 2, the authors use a notation d, I guess as distance. It would not only be helpful to define it but also would be helpful to use a different notation for distance and the d used on page 3, presumably for "empirical state distribution".

11) At the beginning of the paper, the authors motivated the maximum entropy but the final algorithm is based on other approaches.

12) Despite the fact that I could not find this paper ready enough and well-posed, I also have a concern about the novelty of the approach. I think it is not novel enough for publication at ICLR, but I am open to reading other reviewers', as well as commenters', and more especially the authors' rebuttal response.

13) I also encourage the authors to provide a discussion on the cases where the novelty (whatever that could mean) does not matter, rather the  novelty of state-action pair matters.


**Experience Assessment:**

I have published in this field for several years.

**Review Assessment: Checking Correctness Of Derivations And Theory:**

I carefully checked the derivations and theory.

**Review Assessment: Checking Correctness Of Experiments:**

I assessed the sensibility of the experiments.

**Review Assessment: Thoroughness In Paper Reading:**

I read the paper thoroughly.

---

> ### Author Response · Authors · 2019-11-13
> **Replay to Review 1**
>
> Thank you for your detail suggestions.
>
> Q1、Q2、Q3、Q6b、Q7、Q10: Writing styles
>
> Thanks for your suggestion. We have revised our writing styles according to your guidance.
>
> Q4: Definition and notation
>
> $\gamma$ = 0 is excluded since it beyond a standard reinforcement learning (i.e., the current decision considers future rewards). The origin definition of discounted state distribution is singular for $\gamma$ = 1. But we remove this definition since we don’t use it later.
>
> Q5、Q6a: Figure 2
>
> It is not a specific experimental design. It is just an illustration to explain the unexpected results if we try to maximize the empirical state distribution.
>
> Q8: Explanation for the statement
>
> We find the original statement may be incorrect and misunderstanding, thus we rewrite it. We want to illustrate that the chance of discovering new stats is high when performing random actions around states with the least visitation counts rather than other states.
>
> Q9、Q10: Proof of Theorem 2
>
> $\max_{e_t} H_{t+1}$ is the optimization problem of selecting which goals to visit can maximize the entropy. The choices of goals are represented with e_t (i.e., $e_t(i) = 1$ suggest visiting state i).
>
> We have revised the proof of Theorem 2 to make it clear.  We hope the modified version helps you.
>
> Q11: Motivation and method
>
> In fact, we motivate the maximum state entropy exploration helps to find the (near-) optimal policy in reinforcement learning, and we present novelty-pursuit to maximize the state entropy. We assume a perfect goal-conditioned policy and visitation counts oracle in Section 3, but present the practical method in Section 4, verify our method in Section 5.1.
>
> Q12: Novelty of the proposed method
>
> We study the problem of efficient exploration in reinforcement learning. Clearly, current reinforcement learning suffers from suboptimal due to inefficient exploration for environments with huge state space and long horizon. We disclose that goal-conditioned exploration behaviors can also maximize the state entropy, and demonstrate the exploration efficiency. We present the practical methods of such defined goal-conditioned exploration. We appreciate it if you can rethink our work after reading the revised paper and the above responses.
>
> Q13: The novelty of state-action pair matters
>
> We currently only consider the case the state matters. It is known that  $d_\pi(s, a) =  d_\pi(s) * \pi(a|s)$, where $d_\pi(s, a)$ is the state-action distribution and $d_\pi(s)$ is the state distribution. Thus, policy entropy may be helpful in the case of the novelty state-action pair matters. Actually,  our method performs random actions around the exploration boundary, and it may be applicable to that case.

---

### Official Review · AnonReviewer4 · 2019-11-02
**Official Blind Review #4**

**Rating:** 3

**Review:**

This paper proposes novelty-pursuit for exploration in large state space. In theory, novelty-pursuit is motivated by connecting intrinsically motivated goal exploration process (IMGEP) and the maximum state entropy exploration (MSEE), showing that exploring least visited state can increase state distribution entropy most. In practice, novelty-pursuit works in two stages: First, it selects a goal (with largest value prediction error) to train a goal reaching policy to reach the boundary of explored and unexplored states. Second, after reaching goal states, it uses a randomly policy to explore, hopefully can get to unexplored states. Experiments on Empty Room show that the novelty-pursuit with perfect goal reaching policy and visit count information can maximize state distribution entropy. Experiments on Empty Room, Four Rooms, FetchReach and SuperMarioBros show that the proposed method can achieve better performance than vanilla (policy gradient?) and bonus (exploration bonus using Random Network Distillation).

1. The authors claim that the proposed method connects IMGEP and MSEE. However, the theory actually shows that a connection of visit count and MSEE (Thm. 2, choosing least visited state increases the state distribution entropy most.) Table 1 of Empty Room experiments shows the same, with visit count oracle the state entropy is nearly maximized. With goal exploration (entropy 5.35 and 5.47 in Table 1), the state entropy is not "maximized". I consider the theory part and Table 1 more a connection between visit count (including zero visit count, and least visited count) and MSEE, rather than IMGEP and MSEE.

2. The argument of first choose non-visited state, then choose least visited state (Fig. 1) makes sense. However, the experiment design is just one way of approximately achieving this. I did not see why doing this approximation is good from both theoretical and empirical perspectives.

2a) Random Network Distillation (RND) prediction error is used to select goals. After reaching these goals, it is claimed that the boundary of explored and unexplored states has been reached. However, RND just uses visit count as a high-level motivation, and there is no justification that high RND prediction error corresponds to low visit count.

In Table 1, it is surprising even in this simple environment, the entropy still looks not good with approximation. Maybe use larger networks. And why does bonus have the same entropy as random (does not make sense to me since RND should be a much stronger baseline than random policy)?

2b) There exist other methods to approximate this boundary of visited/non-visited states (like pseudo-count as mentioned). Comparisons with other choices are needed (on simple tasks if others cannot be scaled up to SuperMarioBros) to claim that this approximation is a good choice.

3. The experiments are lack of comparison with other exploration methods. There are only comparisons with vanilla (is it policy gradient?) and bonus (I suppose it is exactly the same method as in RND paper?), which is not enough to show the proposed method is on a good level. Also, experiments on more tasks (such as Atari) are needed to evaluate the performance of the purposed method.

4. The reward shaping r(a g_t, g_t) in Eq. (2) is for a changing g_t. In Eq. (7), it seems to show cancelation of fixed g. I did not see why cancelation of fixed g in Eq. (7) can lead to the conclusion that Eq. (2) does not change optimal policies.

Overall, I found this paper: 1) main idea (Fig. 1) makes sense; 2) the theoretical contribution is weak (the connection between visit count and entropy is not difficult to see). It does not connect IMGEP and MSEE, but connects visit count and entropy; 3) The experiments choose one way to approximately reaching boundary of visited and non-visited states, which is lack of comparison with other choices; 4) The experiments look promising, especially on SuperMarioBros, but more experiments on other tasks and comparisons with other exploration methods are needed to evaluate the proposed method thoroughly.

**Experience Assessment:**

I have published one or two papers in this area.

**Review Assessment: Checking Correctness Of Derivations And Theory:**

I carefully checked the derivations and theory.

**Review Assessment: Checking Correctness Of Experiments:**

I carefully checked the experiments.

**Review Assessment: Thoroughness In Paper Reading:**

I read the paper thoroughly.

---

> ### Author Response · Authors · 2019-11-13
> **Replay to Review 4**
>
> Thank you for your helpful review.
>
> Q1: The connection between IMGEP and MSEE
>
> The connection between IMGEP and MSEE is based on goal-conditioned behaviors. We select states with the least visitation counts as goals to maximize state entropy. We consider a perfect goal-conditioned policy and accurate visitation counts in Section 3, thus the practical method seems not to maximize the state entropy shown in Table 1. Note that, with planning-oracles (and a visitation oracle), the gap between our method and the maximum state entropy is only 0.124 (0.039).
>
> Q2a: The method of approximate visitation counts and exploration boundary
>
> First, the origin paper of RND (Burda et a;., 2019) validates that prediction errors given by RND are strongly correlated to the “counts” of training samples on the MNIST dataset. The more samples of a certain class, the lower the prediction errors of that class. Considering we only need the order of states in terms of visitation counts rather the visitation counts itself, RND can meet our needs (See Figure 9 in Appendix 3 for the validation). As for exploration efficiency, we find that the performance improves a little with a visitation counts oracle (See Table 1).
>
> Q2b: Alternative methods to approximate the exploration boundary
>
> As we stated above, the improvement is limited even with visitation counts oracle. Though approximation of visitation counts is not our main study problem, we consider alternative methods (e.g., pseudo-count) in the future works.
>
> Q2b: The performance of novelty-pursuit in Table 1
>
> The gap of our method without oracles and the maximum entropy comes from an imperfect goal-conditioned policy at the early stage of training. Note that training a goal-conditioned policy incurs random actions towards the goal, which decreases the state entropy.
>
> Q2b: The performance of the bonus method in Table 1
>
> The performance of the bonus method does perform well than a random policy. We find that the original result based on 1 seed is unreliable and update the results of Table 1 with 5 seeds. We attribute the limited advantage of the bonus method to delayed and indirect feedback signals of the exploration bonus.
>
> Q3: Compared Baselines
>
> Vanilla policy: DDPG (with Gaussian action noise) on Fetch Reach and ACER (with policy entropy regularization) for others.
> Bonus policy: Off-policy version of RND (origin paper uses the on-policy version) based on vanilla policy. Note that we focus on exploration rather than policy optimization.
> Other baselines and tasks: we think the SuperMarioBros is a better benchmark for deep exploration study and the bonus method is a strong baseline. we are conducting experiments with other baselines in atari.
>
> Q4: Reward Shaping
> First, we do keep the goal unchanged during an episode. For fixed g, the d(ag_{T+1}, g) = 0 and d(ag_1, g_0) is constant. Thus, the optimal policy induced by reward shaping is invariant to the optimal policy induced by Eq. (2). We hope the revised verification in Appendix 2 helps you.

---

### Author Response · Authors · 2019-11-13
**Paper Revision**

 We reivse our writing styles and languages. The updated the paper has the following main changes:

(1) The connection between IMGEP and MSEE is made clear. We revise the languages in Section 3,  simplify notations and update the proof in the Appendix.

(2) We add more details about the exploitation policy in  Section 4.3 to show the difference between ours and the Go-Explore's.

(3) We update the state entropy results reported in Section 5.1 with 5 seeds. We also add Figure 9 in  Appendix A.3 to demonstrate the effectiveness of approximate exploration boundary via prediction errors of RND.

(4) We add trajectories visualization of  SuperMarioBros-1-1 and SuperMarioBros-1-2 in Figure 10 in Appendix A.3.  In particular,  the result of SuperMarioBros-1-1 is used to illustrate that insufficient exploration leads to the local optimum.

---

### Decision · Program_Chairs · 2019-12-19

**Decision:**

Reject

**Comment:**

There is insufficient support to recommend accepting this paper.  The reviewers unanimously recommended rejection, and did not change their recommendation after the author response period.  The technical depth of the paper was criticized, as was the experimental evaluation.  The review comments should help the authors strenghen this work.